# Expert and Novice Teachers’ Cognitive Neural Differences in Understanding Students’ Classroom Action Intentions

**DOI:** 10.3390/brainsci14111080

**Published:** 2024-10-29

**Authors:** Yishan Lin, Rui Li, Jesús Ribosa, David Duran, Binghai Sun

**Affiliations:** 1School of Psychology, Zhejiang Normal University, Jinhua 321004, China; 2Department of Basic, Developmental and Educational Psychology, Universitat Autònoma de Barcelona, 08193 Cerdanyola Del Vallès, Catalonia, Spain

**Keywords:** expert–novice teacher, understanding of intentions, classroom actions, teachers’ professional insight, event-related potential

## Abstract

Objectives: Teachers’ intention understanding ability reflects their professional insight, which is the basis for effective classroom teaching activities. However, the cognitive process and brain mechanism of how teachers understand students’ action intention in class are still unclear. Methods: This study used event-related potential (ERP) technology to explore the cognitive neural differences in intention understanding ability among teachers with different levels of knowledge and experience. The experiment used the comic strips paradigm to examine the ability of expert and novice teachers to understand students’ normative and non-normative classroom actions under different text prompts (“how” and “why”). Results: The results revealed that in the late time window, expert teachers induced larger P300 and LPC amplitudes when they understood students’ classroom action intentions, while the N250 amplitudes induced by novice teachers in the early time window were significantly larger. In addition, for both types of teachers, when understanding the intentions behind students’ normative actions, the N250 amplitude was the most significant, while the P300 and LPC amplitudes were more significant for non-normative actions. Conclusions: This study found that teachers at varying professional development stages had different time processing processes in intention understanding ability, which supported teachers’ brain electrophysiological activities related to social ability.

## 1. Introduction

Teaching expertise includes the application of professional knowledge, problem-solving, pattern recognition, insight, and other cognitive components. Among them, insight, as one of the most important components of teaching expertise, is mainly reflected in the understanding of students’ intentions [1]. Intention understanding refers to a cognitive model in which individuals can observe and understand the psychological state behind others’ actions [2,3]. It consists of four stages: first, identifying actions; second, representing the target behavior; third, understanding the causal relationship between actions and intentions; and finally, generating perceptions and physical re-experiences [4,5,6]. In such a complex environment as the classroom, teachers have the challenge to notice and interpret many different events, which is crucial for effective classroom management [7]. In turn, classroom management has been shown to have an impact on student achievement [8,9,10].

### 1.1. Influence of Expertise Level on Individual Action Recognition

Expertise, broadly defined as the proficiency or skill developed through extensive experience or training in a particular domain, varies significantly across fields [11,12]. In education, teaching expertise goes beyond mere subject knowledge; it encompasses the ability to anticipate, understand, and respond to the dynamic and often unpredictable classroom environment. This includes recognizing patterns in student behaviors and making timely, effective decisions to facilitate learning [13]. Compared to novices, experts demonstrate superior cognitive processing when handling complex information within their area of expertise. They interpret and respond to classroom situations with greater speed and accuracy, relying on their advanced cognitive frameworks and experience. For instance, research in other domains, such as chess, has shown that professional players employ more sophisticated visual search strategies and exhibit enhanced pattern recognition abilities [14,15,16]. Advanced chess players also show heightened alpha EEG (electroencephalogram) activity during complex tasks, compared to novices [17]. Similarly, in tennis, experts display faster and stronger activation of ERP (event-related potential) components during motion perception tasks [18]. However, the cognitive demands of teaching involve additional layers of complexity, including real-time decision making, managing live interactions, and responding to the continuous flow of classroom stimuli.

In the field of teacher education, there are obvious differences between experts and novice teachers in their cognitive processing of classroom teaching situation information [1,19,20,21,22]. Specifically, compared to novices, expert teachers can more quickly identify events in the classroom [20], pay attention to students’ actions earlier [23], and prevent interference from students’ disruptive behavior by identifying behavior and event cues as soon as possible [7,22]. It can be seen that expertise level has a significant impact on teachers’ identification of students’ classroom behaviors. However, current research mainly focuses on revealing the behavioral differences in teachers’ recognition patterns of students’ classroom behaviors, while the cognitive neural processes of more advanced action recognition patterns, that is, how teachers understand students’ action intentions in the classroom, have not been fully elucidated.

### 1.2. Research Paradigm of Action Intention Understanding

The experimental paradigm of action intention understanding is to use specific research designs and methods to deeply study the observer’s cognitive process of the intention behind other people’s actions. In previous studies using the comic strip paradigm, in which subjects were asked to infer relationships and behavioral attributes between two pictures, each trial typically consisted of two pictures; the priming stimulus was an action preparation picture, and the target stimulus was an action execution picture. This design allows researchers to explore how the actions of the person being observed in the picture are performed [24,25,26].

When we observe a target performing an action, we generally receive two pieces of information; one is how the target performs the action, and the other is why the target performs the action [24,25]. For example, based on the action of “brushing teeth”, we can obtain information: why the action is performed (e.g., “clean teeth”) and how to perform it (e.g., “use a toothbrush”). However, previous research focused more on the specific execution methods and paid less attention to the reasons behind the execution of actions. In order to fill this research gap, the current study further expanded the experimental paradigm and aimed to deeply explore the cognitive neural differences in “how” and “why” behaviors are performed to more fully reveal the cognitive process of action intention understanding.

In terms of experimental stimulation, this study used how/why text prompts as the priming stimulus, and students’ classroom actions as the observation objects for the target stimulation. Wang et al. classified not listening carefully as students’ problematic non-normative actions, while behaviors that promoted clear attention were classified as students’ normative actions [21]. At the same time, teachers will subconsciously identify the normativeness of students’ behavior [27]. Therefore, this study subdivided students’ classroom actions into normative and non-normative behaviors to further explore teachers’ understanding of students’ action intentions and the potential impact of different actions on teachers’ intention understanding.

### 1.3. Theoretical Basis and Neural Mechanism of Action Intention Understanding

Embodied cognition theory provides a new perspective for studying the cognitive processes underlying teachers’ initial perception and deep understanding of students’ classroom actions [28,29,30,31]. According to this theory, individuals deepen their understanding of others’ action intentions through both observation and personal experiences [4,6]. This process is particularly relevant in teaching expertise, where teachers utilize their own pedagogical experiences to interpret students’ behaviors, allowing for swift, accurate decision making in real-time classroom environments [32]. Embodied cognition plays a critical role in expertise development, as experts—unlike novices—can draw upon both sensory input and their extensive experience to create richer cognitive representations of observed actions. In teaching, this enables expert teachers to not only interpret students’ behaviors but also to predict potential outcomes and guide their instructional responses accordingly [33]. This highlights the unique demands of teaching expertise, which involves managing live, dynamic interactions and interpreting a continuous flow of stimuli in real time.

From a neurological perspective, the inference of others’ intentions within the framework of embodied cognition can be elucidated through millisecond-level event-related potential (ERP) techniques, which reveal the temporal dynamics of intention understanding. Research shows that the main ERP components related to intention understanding include the N250, P300, and late positive component (LPC) [34]. The N250 visual ERP component produces a more pronounced attention-enhancing effect in the face of familiar stimuli. This means that in the early stages of brain processing, there is a stronger attention-driven response to familiar stimuli to enhance its processing and perception [35]. It has been found that the posterior parietal region of the brain significantly triggers the N250 component 200–250 ms after stimulus presentation, when individuals observe others performing reasonable actions compared to unreasonable actions [36]. P300 reflects the brain’s perception of novel stimuli and processing of negative intentions [37,38]. Huang et al., by using the comic strips paradigm, asked participants to judge the intentions of two characters as friendly cooperative, hostile conflict, and neutral actions in succession [34]. The results showed that 300 ms after stimulus presentation, negative hostile intentions induced a more positive P300 than positive friendly and neutral intentions. LPC appears at a later stage after the stimulus is presented, usually after 400 ms. Studies have shown that LPC is closely related to processes such as emotional processing and social cognition [39]. LPC index is widely used in the task of identifying communicative intention. Wang et al. found that the amplitude of LPC induced by understanding communicative intention in an individual’s brain was significantly larger than that of personal intention, while the amplitude of LPC induced by personal intention was larger than that of physical intention [40]. This neural evidence underscores the role of embodied cognition in understanding others’ intentions, with specific ERP components reflecting distinct stages of cognitive processing during intention understanding.

### 1.4. Study Hypotheses

In order to investigate the behavioral and cognitive neural differences of teachers with different expertise level in understanding students’ classroom action intentions, we recruited expert and novice teachers and linked the “how/why” text prompts with action pictures, so that teachers were asked to infer students’ classroom action intentions, and we used event-related potential technology to explore the neural mechanisms of this process. Hence, we proposed four hypotheses:

**Hypothesis** **1.***Based on the literature review, it is predicted that expert teachers, being more familiar with classroom environments, will have faster reaction times and higher accuracy compared to novice teachers when determining how and why students perform classroom actions. It is also expected that expert teachers will demonstrate greater comprehensibility of students’ classroom behaviors, both before and after the experiment*.

**Hypothesis** **2.***Due to their higher professional expertise, expert teachers are expected to exhibit significantly different ERP results compared to novice teachers, characterized by stronger neural activation in response to understanding classroom action intentions*.

**Hypothesis** **3.***In the early stage of stimulus processing, expert teachers are anticipated to show a stronger N250 amplitude for normative actions compared to non-normative actions, reflecting their familiarity with normative behavior. At later stages of stimulus processing, the amplitudes of P300 and LPC for non-normative actions are expected to be significantly larger than for normative actions, with this effect being more pronounced for expert teachers due to their greater experience in managing classroom behaviors*.

**Hypothesis** **4.***There will be a significant interaction between expertise level, text prompts, action type, and brain regions/hemispheres, with expert teachers exhibiting more distinct neural patterns in response to different classroom action types compared to novice teachers*.

## 2. Materials and Methods

### 2.1. Participants

The study employed a 2 (expertise level: expert vs. novice) × 2 (text prompt: how vs. why) × 2 (action type: normative vs. non-normative) mixed experimental design for the behavioral data, with expertise level as a between-subjects variable and text prompt and action type as within-subjects variables. For the EEG data analysis, we used a 2 (expertise level) × 2 (action type/text prompt) × 3 (brain region) × 3 (hemisphere) design, reflecting the full complexity of the experiment.

Initially, 46 teachers were recruited, but 8 participants were excluded due to large artifacts in their EEG data [41], resulting in a final sample of 38 participants. Power calculations using G*Power 3.1 software, with a more conservative effect size of f = 0.15, indicated a minimum required sample size of 26 participants (13 per group) to achieve adequate power (α = 0.05, power = 0.80; [42,43]). Our final sample of 38 participants provides sufficient power for both behavioral and EEG analyses. All participants were right-handed and had normal or corrected-to-normal vision. The ethics committee approved the study, which was conducted according to the principles outlined in the Declaration of Helsinki.

#### 2.1.1. Expert Teachers

In accordance with previous studies [44,45,46], teachers were classified as experts if they met the following criteria. (1) They were identified by school leaders or educational authorities based on relevant characteristics of expert teachers, and (2) they had at least ten years of teaching experience and were approved by the local government to hold a professional title of Grade I or above. Expert teachers were selected from schools across various cities in Zhejiang Province, China.

In this study, teachers were categorized as experts based on several established criteria utilized by educational authorities. Expert teachers typically exhibit the following characteristics: teaching experience: a minimum of ten years of teaching experience in their subject area; degree requirements: advanced degrees (master’s or higher) in education or related fields; professional development: active participation in ongoing professional development programs; effectiveness in teaching: demonstrated effectiveness in enhancing student learning outcomes, often evaluated through standardized test scores and qualitative assessments from peers and supervisors.

#### 2.1.2. Novice Teachers

Novice teachers were selected according to the screening criteria established by existing national and international studies [45,46]. This group comprised senior college students and newly qualified teachers. Senior college students were enrolled in education-focused universities, had undergone teacher training, and completed over three months of practical experience in schools. Newly qualified teachers were classified as novices based on established criteria set by educational authorities, indicating they had been in the profession for less than five years. The senior students primarily attended normal universities, while the novice teachers were employed in various schools across different cities in Zhejiang Province.

### 2.2. Materials

To enhance ecological validity, the experimental materials were taken with digital cameras. Extant literature has shown that compared with cartoon pictures, photos of real people have higher ecological validity [47,48,49]. The student actor was a male middle school student (age = 15 years, height = 168 cm), and his hairstyle, figure, and dress conformed to the typical characteristics of Chinese students. Students’ normative (e.g., sitting upright, listening attentively, raising their hand to speak) and non-normative (e.g., sleeping, playing on mobile phones, whispering) actions in class were mainly classified according to Ding et al. [50]. The materials included 150 initial photos, half of which were normative actions and half of which were non-normative actions. To exclude differences in participants’ electroencephalographic (EEG) data caused by different physical attributes and task difficulty levels, facial expressions and gaze direction were blurred and the character’s body orientation was matched. All photos were sized to 472 × 354 pixels and matched for luminance, contrast, and color saturation, using Adobe Photoshop 7.0. Twenty teachers (age range: 26–43 years) were selected to evaluate the difficulty and normality degrees of actions (score range: 1–5). No significant differences were found in the difficulty (*p* > 0.05), but a significant difference was identified in the normality (*p* < 0.001). Finally, 70 pictures that met the requirements were selected (normative: 36, non-normative: 34, see Figure 1; specific information on materials is shown in Appendix A).

### 2.3. Subjective Measurements

Before the task, all participants were asked to report their individual teaching-related experience (e.g., teaching experience, professional title, teaching subject) to assess their professional competence, rate the comprehensibility of the student’s classroom actions (1 = incomprehension, 7 = comprehension), and complete the Chinese version of the Interpersonal Reactivity Index (IRI-C; Cronbach’s α of the IRI-C ranges between 0.61 and 0.85 [50]) to evaluate their dispositional empathy, which consists of four dimensions: Empathic Concern (EC), Perspective Taking (PT), Fantasy (FS), and Personal Distress (PD). (1) Empathic Concern (EC) is the tendency of individuals to respond with sympathy and attention to those in distress; (2) Perspective Taking (PT) is the tendency of individuals to take the ideas of others; (3) Fantasy (FS) is the individual’s empathetic response to a fictional character; (4) Personal Distress (PD) is the anxiety and discomfort that individuals experience when they see others suffering [51]. After the task, participants were asked to rate their comprehension of the student’s classroom actions again to compare whether their ability to comprehend the student’s actions improved following the experiment.

### 2.4. Experimental Procedure

All participants were required to complete 8 practice trials before the formal experiment, and 8 action pictures were used as practice materials rather than formal experimental materials. In each trial, a fixation point (“+”) of 500 ms was displayed first to remind participants to concentrate and begin the experiment. Then, a 1500 ms text prompt was presented. At the “how” level, participants needed to consider how the student executed the action in class (e.g., “take out the pen from the pencil case”), and at the “why” level, they needed to consider why the student executed the action in class (e.g., “the reason to take out a pen is to take notes”). Then, participants were shown pictures of normative or non-normative classroom actions for 2000 ms. Participants were asked to press “1” on the keyboard if the classroom action was normative and “2” if it was non-normative. Finally, participants were asked to rate the comprehensibility of the action on a seven-point scale based on the text prompt (1 = incomprehension, 7 = comprehension), and then press the key corresponding to their rating to move to the next trial (Figure 2). The presentation order was pseudo-random, with a total of four blocks in the formal experiments, with 64 trials under each block (32 trials each for normative and non-normative actions). The entire experiment lasted 30–40 min; the study used E-prime 3.0 for programming. A Dell LED computer monitor (13.3 inches, 60 Hz refresh rate, 2560 × 1600 resolution) was used. Participants were 50 cm away from the screen.

### 2.5. Electrophysiological Recording and Analysis

EEG data were recorded using the 64-channel Brain Product equipment according to the extended international 10–20 system. The sampling frequency was 500 Hz, AC acquisition was adopted, the filter bandpass was 0.1~100 Hz, and the impedance between all electrodes and scalp was maintained below 5 kΩ. During EEG recording, all electrodes used FCz as the reference electrode. During offline analysis, they were converted to whole brain averages for re-reference. Offline signal processing was performed using EEGLAB 13.0 and the ERPLAB Toolbox v10.04 [52,53]. A filter with a bandpass of 0.1~30 Hz was used to remove high-frequency noise. Independent component analysis (ICA) was used to reject blinks and eye movement artifacts. All trials in which EEG voltages exceeded a threshold of ±75 μV were excluded from the analysis. EEG data were segmented in epochs from 200 ms before to 1000 ms after stimulus onset. The moments when the picture of the intended level of the starting stimulus and picture of the action type of the target stimulus appeared were marked. According to the experimental conditions, the ERP waveform was superimposed to obtain the average total waveform of each participant.

Based on previous research on action intention understanding [34,40,49,54], we conducted an observation and analysis of the average EEG data across all trials. From this analysis, we selected nine electrode sites for further examination: F3, Fz, F4, C3, Cz, C4, P3, Pz, and P4. The time window of the N250 wave was 170–270 ms after stimulation, the time window of the P300 wave was 270–450 ms, and the LPC time wave window was 450–750 ms. Since early components are relatively sharp and late components are relatively gentle, early components were measured by the peak value and LPC was measured by average amplitude.

We utilized SPSS 22.0 to perform repeated measures ANOVAs with a 2 (expertise level: expert vs. novice) × 2 (text prompt: how vs. why) × 2 (action type: normative vs. non-normative) design. The dependent variables included response time, accuracy, and comprehensibility for the behavioral data. For EEG data, to avoid false positive results, we analyzed the EEG data in two parts, and we excluded trials with errors from the EEG data to maintain the quality and accuracy of the results. In the first part, a 2 × 2 × 3 × 3 repeated measures analysis of variance (ANOVA) was conducted on N250, P300, and LPC, with expertise level as the between-subjects factor variable, and action type, brain region, and hemisphere as within-subjects factor variables. In the second part, a 2 × 2 × 3 × 3 repeated measures ANOVA was performed on N250, P300, and LPC, with expertise level as the between-subjects factor variable, and text prompt, brain region, and hemisphere as within-subjects factor variables. The Greenhouse–Geisser correction was applied to adjust the degrees of freedom when the assumptions of sphericity were violated. Post hoc comparisons were conducted using ANOVA with a False Discovery Rate (FDR) adjustment based on the adjusted *p*-values.

## 3. Results

### 3.1. Subjective Measurements

The independent samples *t*-test results showed significant differences between the expert and novice teachers in the three dimensions of the IRI-C scale (Table 1); expert teachers scored significantly higher than novice teachers in perspective taking (PT), fantasy (FS), and empathy concern (EC; *p*s < 0.001), but no significant difference was found in personal distress (PD) (*p* = 0.500). A two-factor repeated measures ANOVA was conducted on expertise level and comprehension of students’ classroom actions before and after the experiment. The results showed that the main effects of expertise level and test time were significant (*p*s < 0.05). Expert teachers scored significantly higher than novice teachers for comprehensibility of students’ classroom actions, and the degree of comprehensibility of students’ classroom actions was significantly higher after the experiment compared with before.

The interaction between expertise level and test time was not significant, F(1, 36) = 0.019, *p* = 0.890, *η*_p_^2^ = 0.001 (Table 2). In summary, the subjective measurement report results support part of Hypothesis 1, that is, expert teachers have a higher comprehensibility of students’ classroom action before and after the experiment than novice teachers.

### 3.2. Behavioral Results

Table 3 shows the results for reaction time, accuracy, and comprehensibility of text prompts on actions. For reaction time, the main effect of text prompt was significant, F(1, 36) = 4.93, *p* < 0.05, *η*_p_^2^ = 0.12, with reactions being faster at the “how” level (733 ± 195.47 ms) than the “why” level (756.96 ± 209.07 ms). Regarding accuracy rate, the results showed a significant main effect of action type, F(1, 36) = 13.17, *p* < 0.01, *η*_p_^2^ = 0.27, with the accuracy rate being higher for non-normative actions (95.89 ± 5.31%), compared with normative actions (87.64 ± 12.02%). The interaction between text prompt and action type was significant, F(1, 36) = 5.76, *p* < 0.05, *η*_p_^2^ = 0.14. Simple effects analysis showed that the accuracy rate was higher at the “how” level (89.06 ± 1.83%) than at the “why” level (86.23 ± 2.10%) for normative actions. For non-normative actions, the difference between levels was not significant (*p* = 0.260). The comprehensibility of text prompts on actions results showed that the main effect of text prompt was significant, F(1, 36) = 11.18, *p* < 0.01, *η*_p_^2^ = 0.24 (Table 3), with comprehensibility being higher at the “how” level (6.08 ± 0.96) than the “why” level (5.93 ± 1.05). The main effect of action type was significant, F(1, 36) = 22.48, *p* < 0.001, *η*_p_^2^ = 0.38, with comprehensibility being higher for normative actions (6.45 ± 0.55) than for non-normative actions (5.56 ± 1.15). The main effect of expertise level is not significant, F(1, 36) = 2.133, *p* = 0.153, *η*_p_^2^ = 0.056. The three-factor interaction is not significant (*p*s > 0.05).

In contrast, these results do not support Hypothesis 1, that expert and novice teachers differ in response time and accuracy. The reason for this result may be that during the experimental procedures, both types of teachers combined text prompts to form rapid explicit behavior judgments on students’ actions. Therefore, it is necessary to further examine the time processing process of the two types of teachers’ intention understanding based on EEG results.

### 3.3. Electrophysiological Results

To examine the effects of expertise level and action type on EEG activity, we conducted a 2 × 2 × 3 × 3 repeated measures analysis of variance (ANOVA) on the N250, P300, and LPC components. Expertise level was treated as the between-subjects factor, while action type, brain region, and hemisphere were included as within-subject factors. The specific results are as follows.

#### 3.3.1. N250 Component

As shown in Figure 3, the N250 component was observed in the 170–270 ms time window. Novice teachers exhibited significantly larger N250 amplitudes (0.36 ± 0.22 μV) compared to expert teachers (1.65 ± 0.22 μV), with a main effect of expertise level, F(1, 36) = 17.23, *p* < 0.001, *η*_p_^2^ = 0.32. Normative actions also induced larger N250 amplitudes (0.92 ± 0.16 μV) than non-normative actions (1.90 ± 0.16 μV), with a main effect of action type, F(1, 36) = 4.70, *p* = 0.037, *η*_p_^2^ = 0.12.

Significant interactions between expertise level and brain region (F(2, 72) = 3.70, *p* = 0.300, *η*_p_^2^ = 0.09) showed that novice teachers elicited more negative N250 amplitudes in frontal and central regions compared to expert teachers (*p*s < 0.001). Furthermore, the interaction of action type and brain region (F(2, 72) = 4.08, *p* = 0.021, *η*_p_^2^ = 0.10) and action type and brain hemisphere (F(2, 72) = 7.36, *p* < 0.001, *η*_p_^2^ = 0.17) were significant. In the frontal and central region, normative actions induced a more negative N250 amplitude than non-normative actions (*p*s < 0.001). Normative actions elicited larger N250 amplitudes in the left hemisphere and central line (*p*s < 0.001), while non-normative actions induced stronger amplitudes in the right hemisphere (*p* < 0.05).

#### 3.3.2. P300 Component

As shown in Figure 3, the P300 component was observed within the 270–450 ms time window. Expert teachers exhibited significantly larger P300 amplitudes (1.63 ± 0.21 μV) compared to novice teachers (0.64 ± 0.21 μV), demonstrating a main effect of expertise level, F(1, 36) = 11.37, *p* < 0.001, *η*_p_^2^ = 0.24. Additionally, non-normative actions resulted in larger P300 amplitudes (1.26 ± 0.16 μV) than normative actions (1.01 ± 0.15 μV), with a main effect of action type, F(1, 36) = 8.94, *p* = 0.005, *η*_p_^2^ = 0.20.

Significant interactions were observed between expertise level and brain region, F(2, 72) = 4.21, *p* < 0.05, *η*_p_^2^ = 0.11; simple effects analysis showed that expert teachers elicited larger P300 amplitudes in the frontal and central regions compared to novice teachers (*p*s < 0.01). Furthermore, the interactions of action type and brain region (F(2, 72) = 10.92, *p* < 0.001, *η*_p_^2^ = 0.23) and action type and brain hemisphere (F(2, 72) = 21.69, *p* < 0.001, *η*_p_^2^ = 0.38) were also significant. Specifically, in the frontal and central regions, non-normative actions induced larger P300 amplitudes compared to normative actions (*p*s < 0.001). Non-normative actions elicited larger P300 amplitudes in the left hemisphere and along the central line (*p*s < 0.001), while normative actions resulted in stronger amplitudes in the right hemisphere (*p* < 0.05). Interestingly, the interaction of action type, brain hemisphere, and brain region was significant, F(4, 144) = 3.34, *p* = 0.012, *η*_p_^2^ = 0.09. At a certain electrode points (F3, Fz, C3, Cz), non-normative actions induced larger P300 amplitudes than normative actions (*p*s < 0.001), while at C4 and P4, normative actions induced larger P300 amplitudes than non-normative actions (*p*s < 0.001).

#### 3.3.3. Late Positive Component

As shown in Figure 3, LPC was observed within the 270–450 ms time window. Expert teachers exhibited significantly larger LPC amplitudes (1.07 ± 0.20 µV) compared to novice teachers (0.50 ± 0.20 µV), demonstrating a main effect of expertise level, F(1, 36) = 3.85, *p* = 0.048, *η*_p_^2^ = 0.097. Additionally, non-normative actions resulted in larger LPC amplitudes (0.99 ± 0.16 μV) than normative actions (0.59 ± 0.15 μV), with a main effect of action type, F(1, 36) = 14.10, *p* = 0.001, *η*_p_^2^ = 0.28.

Significant interactions were observed between expertise level and action type, F (1, 36) = 3.92, *p* = 0.045, *η*_p_^2^ = 0.105; simple effects analysis showed that expert teachers elicited larger LPC amplitudes in the normative action condition compared to novice teachers (*p* < 0.05). Furthermore, the interactions of action type and brain region (F(2, 72) = 8.86, *p* < 0.001, *η*_p_^2^ = 0.20) and action type and brain hemisphere (F(2, 72) = 18.67, *p* < 0.001, *η*_p_^2^ = 0.34) were also significant. Specifically, in the frontal and central regions, non-normative actions induced larger LPC amplitudes compared to normative actions (*p*s < 0.001). Non-normative actions elicited larger LPC amplitudes in the left hemisphere, center line and right hemisphere (*p*s < 0.05). Surprisingly, the interaction of action type, brain hemisphere, and brain region was significant, F(4, 144) = 3.83, *p* = 0.005, *η*_p_^2^ = 0.10. At certain electrode points (F3, Fz, C3, Cz), non-normative actions induced larger LPC amplitudes than normative actions (*p*s < 0.001), while at P4, normative actions induced larger LPC amplitudes than non-normative actions (*p*s < 0.001). Consistent with the previous P300 amplitude, in the left fronto-central area of the brain, especially at the F3, Fz, C3, and Cz electrode points, the LPC amplitude of expert teachers was more positive than that of novice teachers. The LPC amplitude of teachers’ understanding of students’ intentions for non-normative actions was more positive than that for normative actions.

Overall, the ANOVA analysis of the first four factors supports Hypotheses 2, 3, and partially Hypothesis 4. Significant differences in ERP results were found between expert and novice teachers when interpreting students’ classroom action intentions. Specifically, normative actions elicited a more pronounced N250 amplitude than non-normative actions, while non-normative actions led to larger P300 and LPC amplitudes compared to normative ones. Moreover, there was a significant interaction between expertise level, action type, and brain region.

To examine the effects of expertise level and text prompts on EEG activity, we conducted a second 2 × 2 × 3 × 3 repeated measures analysis of variance (ANOVA) on the N250, P300, and LPC components. Expertise level was treated as the between-subjects factor, while text prompts, brain region, and hemisphere were included as within-subject factors. The specific results are as follows.

#### 3.3.4. N250 (A Second Repeated Measures ANOVA)

The results, as shown in Figure 4, indicate that the main effects of expertise level and text prompts were not significant (*p*s > 0.05). However, there was a significant interaction between expertise level and brain region, F(2, 72) = 8.85, *p* < 0.001, *η*_p_^2^ = 0.20. Simple effects analysis revealed that novice teachers exhibited a larger N250 amplitude in the frontal region compared to expert teachers (*p* < 0.001), while expert teachers showed a larger N250 amplitude in the parietal region (*p* < 0.05), with no significant differences in the central region. Moreover, the interaction of expertise level, brain region, and brain hemisphere was significant, F(4, 144) = 3.04, *p* = 0.19, *η*_p_^2^ = 0.08. In the frontal lobes, at the F3 and F4 electrode points, novice teachers had a more negative N250 amplitude than expert teachers (*p*s < 0.05), while expert teachers had a larger N250 amplitude at the P3 and P4 electrode points (*p*s < 0.05). Like the first four-factor ANOVA analysis, whether text prompts are used as priming stimuli or student action types are used as target stimuli, in the frontal lobes of the brain (especially at the F3 and F4 electrode points), it was found that teachers’ expertise levels yielded differences in understanding students’ classroom action intention, and the N250 amplitude of novice teachers was more negative than that of expert teachers.

#### 3.3.5. P300 (A Second Repeated Measures ANOVA)

As shown in Figure 4, the results indicated a significant main effect of text prompts, F(1, 36) = 5.76, *p* = 0.022, *η*_p_^2^ = 0.13. The “why” prompt (0.93 ± 0.12 μV) elicited larger P300 amplitudes than the “how” prompt (0.73 ± 0.13 μV; *p* < 0.01). There was also a significant interaction between text prompt, brain region, and hemisphere, F(2, 72) = 6.21, *p* < 0.001, *η*_p_^2^ = 0.15. Simple effects analysis showed that the “why” prompt induced larger P300 amplitudes at the F3, C3, and P3 electrodes compared to the “how” prompt (*p*s < 0.05), while at the F4 electrode, the “how” prompt induced a larger P300 amplitude than the “why” prompt (*p* < 0.001). Similar to the findings from the four-factor ANOVA analysis, in the left hemisphere, particularly at the F3, C3, and P3 electrode points, the P300 amplitude for understanding why students perform certain actions was more positive than for understanding how they perform those actions.

#### 3.3.6. LPC (A Second Repeated Measures ANOVA)

As shown in Figure 5, the results revealed a significant main effect of expertise level, F(1, 36) = 4.41, *p* = 0.043, *η*_p_^2^ = 0.11, where expert teachers (1.07 ± 0.13 μV) exhibited larger LPC amplitudes than novice teachers (−0.23 ± 0.13 μV; *p* < 0.05). The interaction between expertise level, brain region, and hemisphere was also significant, F(4, 144) = 2.72, *p* = 0.032, *η*_p_^2^ = 0.07. Simple effects analysis showed that expert teachers induced larger LPC amplitudes at electrode sites F3, Fz, C3, and P3 compared to novice teachers (*p*s < 0.05).

Consistent with the results of the four-factor ANOVA analysis, regardless of whether text prompts or student action types were employed, expert teachers consistently exhibited more positive LPC amplitudes than novice teachers in the left fronto-central brain region, particularly at the F3, Fz, C3, and P3 electrode sites. In summary, the second four-factor ANOVA analysis partially confirmed Hypothesis 4, revealing a significant interaction between expertise level, text prompts, and neural activity in teachers’ brains.

#### 3.3.7. Topographical Map

This study focuses on the differences between expert and novice teachers in understanding normative and non-normative classroom actions, and this difference is confirmed by brain topography (Figure 6). On the one hand, there is a trend of stepwise activation enhancement in the 170–270 ms, 270–450 ms, and 450–750 ms time windows, indicating that the teacher understands the student’s intention to a larger degree at late activation. On the other hand, expert teachers have larger brain activation than novice teachers in the 270–450 ms and 450–750 ms time windows, which is the same as the ERP results.

## 4. Discussion

This study used the comic strips paradigm to explore the behavioral and cognitive neural differences in intention understanding between expert and novice teachers. Our subjective measurement results show that expert teachers have a significantly higher comprehensibility of student classroom action before and after the experiment than novice teachers, and score higher on PT, FS, and ES on the IRI-C scale; behavioral results show that expert teachers and novice teachers had no significant differences in reaction time and accuracy rate, but there were significant differences in ERP results. Specifically, expert teachers showed larger P300 and LPC in the later stages of classroom action-type processing, while novice teachers showed larger N250 in the early stages. Furthermore, students’ normative actions induced more negative N250 amplitudes, whereas non-normative actions induced more positive P300 and LPC amplitudes. Teachers must engage in many cognitive activities to guide student learning [55], and professional development helps identify meaningful patterns in the classroom, which in turn enables teachers to improve interactions with students [56]. The results of this study further extend this idea, and teachers must not only be able to distinguish and identify students’ actions in the classroom, but also understand the intentions and reasons behind them.

Expert teachers consistently scored higher than novice teachers on perspective taking, fantasy, and empathy concern, in line with previous research showing that expert teachers possess higher empathy and cognitive flexibility due to their extensive teaching experience [57]. This study also found that expert teachers had a better understanding of students’ classroom actions post-experiment, corroborating earlier findings that expert teachers possess richer classroom knowledge, enabling them to process complex information more efficiently and recognize meaningful patterns in classroom interactions [58]. However, contrary to expectations, no significant differences in behavioral performance (reaction time and accuracy) were found between expert and novice teachers. This lack of a significant behavioral difference could be attributed to the specific structure of the EEG task, which required rapid judgments based on a 1500 ms text prompt followed by a 2000 ms student action stimulus. This short timeframe for processing may have masked differences in explicit behavior performance, despite the underlying cognitive differences revealed by the ERP data.

ERP results indicate that novice teachers exhibited larger N250 amplitudes in response to classroom actions, particularly in the fronto-central regions of the brain. The N250 component is linked to the brain’s predictive processing system [59], where larger amplitudes are indicative of greater prediction error or heightened arousal when encountering unfamiliar stimuli. Novice teachers, due to their relative inexperience, are more likely to experience discrepancies between expected and actual classroom actions, resulting in the larger N250 amplitudes. This supports the notion that novice teachers may face greater cognitive challenges in interpreting students’ behaviors, as they must allocate more cognitive resources toward making sense of classroom interactions. In contrast, normative actions—those that align with teachers’ expectations—elicited more negative N250 amplitudes, indicating that familiar behaviors are processed with less cognitive effort. This mirrors findings from studies on face recognition [60,61,62], where familiar faces elicit larger N250 responses due to their ability to attract and hold attention more effectively.

Expert teachers, on the other hand, showed larger P300 and LPC amplitudes in the later stages of processing students’ actions (270–750 ms post-stimulus). These components are associated with deeper cognitive processing and the integration of knowledge. The P300 amplitude, in particular, is modulated by prior knowledge and experience [63]. In this study, expert teachers, with their rich classroom experience, demonstrated greater cognitive activation when processing student behaviors, especially non-normative actions. These unexpected or non-standard classroom behaviors require more attention and cognitive resources to interpret, explaining the larger P300 amplitudes [18]. The LPC, reflecting higher-order social cognitive processes such as theory of mind [40], also showed larger amplitudes in expert teachers, indicating that they engage in more complex cognitive evaluations of students’ actions. This aligns with previous research suggesting that expert teachers process classroom interactions at a deeper level [7], utilizing their knowledge and experience to make sense of students’ intentions. Non-normative student behaviors—those that do not conform to classroom expectations—elicited larger P300 and LPC amplitudes than normative actions. The P300 component is sensitive to unexpected events, and larger amplitudes are observed when individuals encounter stimuli that violate their expectations [64,65,66]. In classroom settings, non-normative behaviors require teachers to expend more cognitive effort to understand students’ underlying intentions, particularly those of more active or disruptive students, as opposed to quieter ones [67]. This deeper level of cognitive engagement is reflected in the larger P300 and LPC responses in expert teachers.

The behavioral results indicate that teachers’ response times to “how” prompts are significantly faster than those for “why” prompts, with higher accuracy in their judgments regarding students’ action intentions at the “how” level. This finding suggests that lower-level prompts are easier to process. In contrast, the EEG results reveal that the P300 amplitude induced by “why” prompts is significantly greater than that induced by “how” prompts. Research indicated that understanding the intentions behind “why” questions primarily activates brain regions associated with theory of mind, such as the ventromedial prefrontal cortex [68]. This processing is influenced by cognitive load, whereas understanding “how” predominantly engages areas related to the mirror neuron system, such as the inferior parietal lobule, which is less affected by cognitive demands. In this experiment, understanding “how” requires teachers to recognize students’ actions at a basic level, while comprehending “why” students engage in certain behaviors necessitates deeper cognitive processing, involving critical thinking about the underlying motivations for those actions. Consequently, grasping the reasons behind students’ behaviors demands additional cognitive effort, thereby consuming more cognitive resources from teachers.

### Future Research

Our study offers some promising preliminary findings on the potential participation of the brain in the social capability of expert and novice teachers. Our results also suggest future research directions. Firstly, given the improvement in teachers’ perceived comprehensibility of students’ classroom actions and dispositional empathy, future studies should plan and analyze the effect of interventions aimed at strengthening the training of novice teachers’ social capacity. They might be provided with practical opportunities to work on their perceptions and interpretations of, and responses to, classroom information, as well as their understanding of classroom teaching to flexibly respond to various uncertainties. Approaches such as reflective practice (e.g., [69]) and teacher collaboration (e.g., [70]) might become fertile ground for future research on the topic in relation to teacher professional development. Secondly, from the perspective of emotion and motivation, future research might provide new insights into the mechanism of teachers’ intention understanding ability. Thirdly, considering that Berliner points to cultural differences in the criteria that are set to define an expert teacher, cross-cultural similarities and differences might be addressed in subsequent studies [58]. Fourthly, future research should consider if the findings of this study about the intention understanding ability of expert teachers are extended to (and related to) other teaching expertise areas (e.g., whether expert teachers’ selective attention and knowledge-based reasoning in professional insight will have the same impact on the brain). Overall, expanding the educational neuroscience research on teachers’ social abilities might help to improve teaching practice. In the future, studies in experimental and natural contexts should complement each other to ensure the ecological validity of the findings, especially considering that the interpretation of normative and non-normative actions might be context-dependent.

## 5. Conclusions

This study combined behavioral and ERP techniques to explore the dynamic processing time course of teachers’ brains in understanding students’ classroom action intentions. We found that, compared with novices, expert teachers demonstrated different patterns of brain activity, as reflected in the P300 and LPC amplitudes during the late time window. Additionally, novice teachers exhibited a stronger N250 amplitude in the early time window, indicating a potentially heightened response to normative actions. Notably, the N250 amplitude was more pronounced for normative actions, whereas the P300 and LPC amplitudes were significantly larger for non-normative actions. From the perspective of cognitive neuroscience, this study shows that expert teachers produce late and sustained processing in understanding students’ classroom action intentions, and text prompts and action types are important indicators that affect teachers’ understanding of student classroom action intentions.

## Figures and Tables

**Figure 1 brainsci-14-01080-f001:**
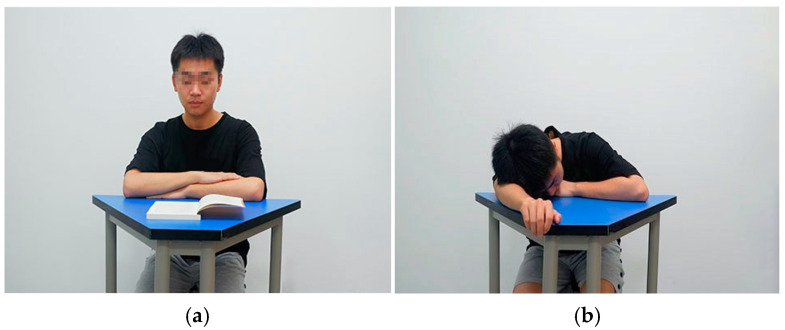
Examples of experimental materials: (**a**) normative action; (**b**) non-normative action.

**Figure 2 brainsci-14-01080-f002:**
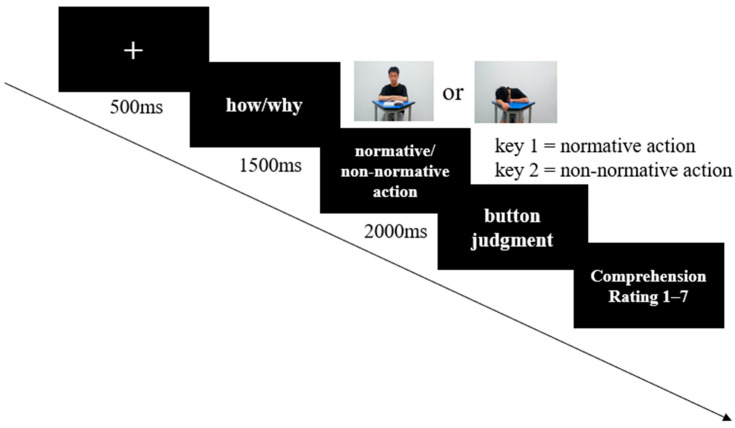
Experimental procedure used in the study. Each trial consisted of a specific sequence: a fixation point was presented for 500 ms, followed by either a “how” or “why” text prompt, which was displayed for 1500 ms in a random order across trials. Subsequently, photographs of students engaged in normative and non-normative actions were presented for 2000 ms. After viewing the photographs, participants made a key press judgment regarding the actions and then rated the comprehensibility of both the text prompt and the student photographs.

**Figure 3 brainsci-14-01080-f003:**
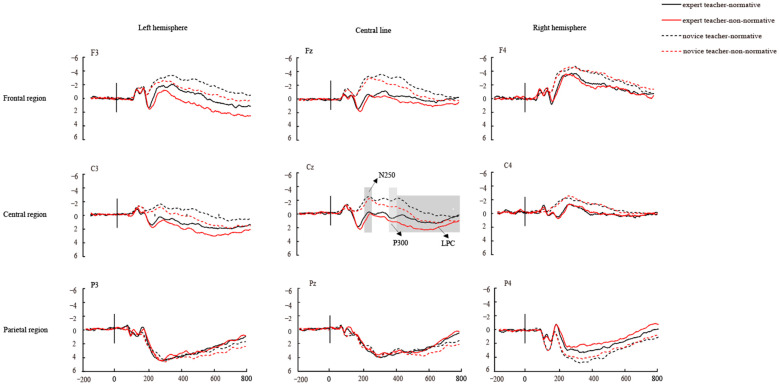
Expertise level and action type of differential wave in different electrode points.

**Figure 4 brainsci-14-01080-f004:**
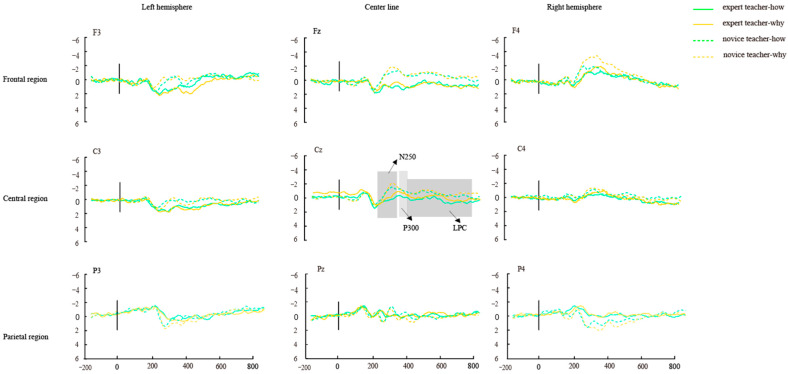
Expertise level and text prompt of differential wave in different electrode points.

**Figure 5 brainsci-14-01080-f005:**
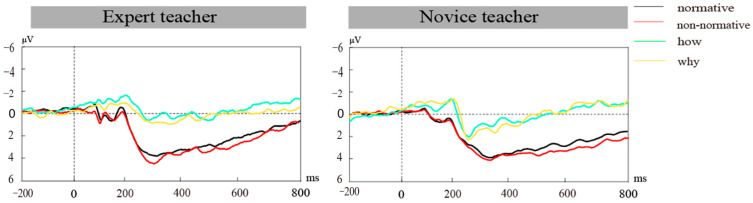
Action type and text prompt of differential wave in different expertise levels.

**Figure 6 brainsci-14-01080-f006:**
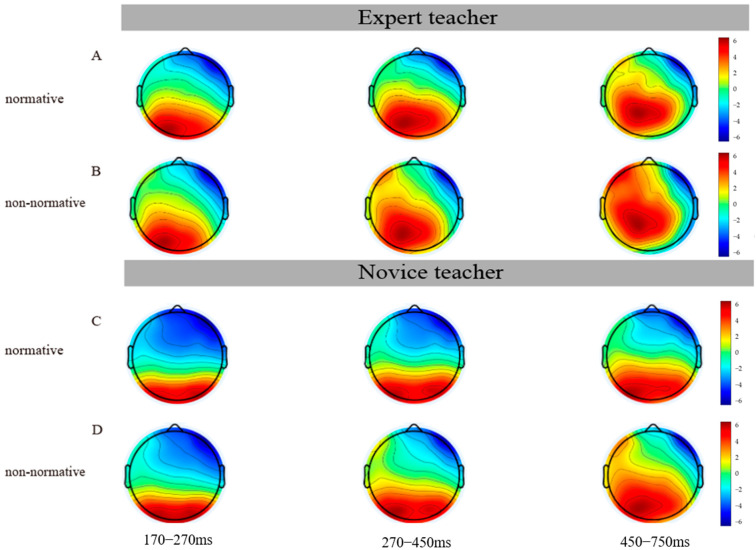
Topographical maps of expertise level and action type in N250, P300, and LPC. (**A**) Topographic map of expert teacher under normative action, (**B**) Topographic map of expert teacher under non-normative action, (**C**) Topographic map of novice teacher under normative action, (**D**) Topographic map of novice teacher under non-normative action.

**Table 1 brainsci-14-01080-t001:** Descriptive statistics for psychological measurements (M ± SD).

	Expert Teacher	Novice Teacher
	M ± SD	M ± SD
PT ^1^	21.05 ± 2.272	17.89 ± 2.923
FS	22.32 ± 3.845	17.84 ± 2.794
EC	24.37 ± 3.515	18.47 ± 1.775
PD	15.74 ± 4.012	14.89 ± 3.588
Pre-test	5.68 ± 0.946	4.74 ± 0.991
Post-test	6.11 ± 0.737	5.11 ± 1.049

^1^ Pre- and Post-test in the comprehensibility of classroom action. PT: perspective taking; FS: fantasy; EC: empathy concern; PD: personal distress.

**Table 2 brainsci-14-01080-t002:** Two-factor repeated measures ANOVA for expertise level and test time.

	SS	df	MS	F	*p*	*η* _p_ ^2^
Expertise level	18.02	1	18.02	16.70	0.001 ***	0.32
Test time	2.96	1	2.96	4.34	0.044 *	0.11
Expertise level × Test time	0.01	1	0.01	0.02	0.894	0.001

* means *p* < 0.05, the difference is significant; *** means *p* < 0.001, the difference is extremely significant.

**Table 3 brainsci-14-01080-t003:** Reaction time (ms), accuracy (%), and comprehensibility results of two groups.

	Expert (M ± SD)		Novice (M ± SD)	
	Normative	Non-Normative	Normative	Non-Normative
how-RT	717.03 *±* 145.69	768.39 *±* 145.74	722.98 *±* 232.18	723.60 *±* 247.07
why-RT	754.28 *±* 160.02	774.69 *±* 144.69	744.68 *±* 258.22	754.19 *±* 261.54
how-ACC	88.08 *±* 13.28	83.34 *±* 7.00	90.05 *±* 8.78	97.70 *±* 2.29
why-ACC	84.95 *±* 12.23	95.07 *±* 6.32	87.50 *±* 13.59	97.45 *±* 3.13
how-comprehensibility	6.67 ± 0.43	5.76 ± 0.94	6.37 ± 0.53	5.52 ± 1.27
why-comprehensibility	6.56 ± 0.52	5.67 ± 1.02	6.21 ± 0.64	5.29 ± 1.36

## Data Availability

The data provided in this study are available upon request from the corresponding authors due to privacy and ethical considerations. As the data involve sensitive personal information related to participants, we have implemented this restriction to protect their privacy and comply with ethical standards.

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
