# Peer review of "Expert and Novice Teachers’ Cognitive Neural Differences in Understanding Students’ Classroom Action Intentions"

_brainsci, 2024, doi:10.3390/brainsci14111080_

Round 1
Reviewer 1 Report
Comments and Suggestions for Authors
the manuscript "Expert and Novice Teachers’ Cognitive Neural Differences in Understanding Students’ Classroom Action Intentions" is an EEG study comparing expert and novice teacher by using a comic strip paradigm but with photos from a real student (15 y old).
The manuscript is very dense and has a few weaknesses that can be fixed
1. the introduction would benefit from a re-ordering, i.e. the how/why before the specifics of the N250 and P300.
2. the hypotheses should be direcitonal were possible, i.e. from the literature review one would predict that expert teachers are faster (H1)
and since expert teachers are more familiar with normative behaviour show a strogner N250 ...
3. since there are 3 outcomes (in the repeated measures ANOVAS) and 2 versions (action or text prompt) the p-value should be adjusted, too restrictive would be to use Bonferoni correction (p < .0083), FDR or FWE will do too.
4. where did you get the effect size from, line 163? This is a rather big effect, please justify it. Also; is iths ifr a 2x2 design? Or a 2x3x3 design? Furthermore, you run multiple analysis, and it should be based on the most complex one, here the 2x2x3x3
5. analysis: were trials with error excluded from ERP / EEG analysis?
6. discussion: larger N250 in novice teachers may indicate that they are less familiar with the behaviour, hence larger prediction error or arousal of seeing that, that fits with the predictive processing account of the brain
7. the discussion is devoid of the how/why difference?
8. conclusion: "... better understood student's behavioural intentions..." - even if you see a difference in P300 and LPC you cannot claim that. You have not measured it, this is inferred solely from the LPC and P300 but there are alternative explaantions. Maybe yoou can say so if the P300 and LPC is higher in why than how trials of non-normative behaviour, but you will not have the stastical power for a 2x2x2 analysis (expert/novice x how/why x normative/non-normative) and LPC and P300 as outcome.
minor: please report always exact p values, not p<.05 but p = .246, also the significant p-value should be written out (unless smaller than .001)
line 178 what / where is "normal university"?
please expand the legend for figure 2. normative /non-normative action is a photo? how/why is separate trials or always shown both? (from later mentioned text it seems it is separate trials)
line 277: tabel should be table
line 291 space issue .001(Table 2) should be .001 (Table 2)
line 442/443 / 455: 2x simple, delete one instance
figure 4 and 5 are not shwon properly, the legend is out of range
line 473: "... larger degree of late activation" should that be "larger degree at late activation"?
line 475 space issue the270
there are general space issues for F() i.e. there should be no space between F and bracket
line 484ff: "... show that ... novel teachers there were no ..." grammatically wrong. replace "there were no" with "had no"
line 516: trails should be trials
overall an interesting read, thanks.
Comments on the Quality of English Languagesome grammatically odd construct
e.g. line 433ff "... teacher's expertise levels have differences in understanding students' classroom action intention,..."
could be "yielded differences" but since there is no group differences at the behavioural level, the entire construct doesn't work
there are similar instances throught the manuscript
another example on line 349ff: ...teachers understand that the N250 amplitude ..." I don't think teachers understand the N250 amplitude, please rewrite
Author Response
the manuscript "Expert and Novice Teachers’ Cognitive Neural Differences in Understanding Students’ Classroom Action Intentions" is an EEG study comparing expert and novice teacher by using a comic strip paradigm but with photos from a real student (15 y old).
The manuscript is very dense and has a few weaknesses that can be fixed
- the introduction would benefit from a re-ordering, i.e. the how/why before the specifics of the N250 and P300.
Response: Thank you for your valuable suggestion regarding the structure of the introduction. We have restructured it according to your recommendation. The introduction now begins with a broader explanation of the how and why aspects of action understanding, followed by a discussion of the specific ERP components (N250, P300, and LPC) and their relevance to intention understanding. Please see sections 1.2. "Research Paradigm of Action Intention Understanding" and 1.3. "Theoretical Basis and Neural Mechanism of Action Intention Understanding."
- the hypotheses should be direcitonal were possible, i.e. from the literature review one would predict that expert teachers are faster (H1)
and since expert teachers are more familiar with normative behaviour show a strogner N250 ...
Response: Thank you for your insightful feedback regarding the directionality of our hypotheses. We have revised our hypotheses to incorporate more specific predictions based on the literature review. Please see the revised hypotheses section in the introduction, specifically lines 145-161 highlighted in blue.
“Hypothesis 1: Based on the literature review, it is predicted that expert teachers, being more familiar with classroom environments, will have faster reaction times and higher accuracy compared to novice teachers when determining how and why students perform classroom actions. It is also expected that expert teachers will demonstrate greater comprehensibility of students' classroom behaviors both before and after the experiment.
Hypothesis 2: Due to their higher professional expertise, expert teachers are expected to exhibit significantly different ERP results compared to novice teachers, characterized by stronger neural activation in response to understanding classroom action intentions.
Hypothesis 3: In the early stage of stimulus processing, expert teachers are anticipated to show a stronger N250 amplitude for normative actions compared to non-normative actions, reflecting their familiarity with normative behavior. At later stages of stimulus processing, the amplitudes of P300 and LPC for non-normative actions are expected to be significantly larger than for normative actions, with this effect being more pronounced for expert teachers due to their greater experience in managing classroom behaviors.
Hypothesis 4: There will be a significant interaction between expertise level, text prompts, action type, and brain regions/hemispheres, with expert teachers exhibiting more distinct neural patterns in response to different classroom action types compared to novice teachers.”.
- since there are 3 outcomes (in the repeated measures ANOVAS) and 2 versions (action or text prompt) the p-value should be adjusted, too restrictive would be to use Bonferoni correction (p < .0083), FDR or FWE will do too.
Response: Thank you for your insightful feedback regarding the adjustment of p-values in our repeated measures ANOVAs. We appreciate your concern about the potential stringency of the Bonferroni correction (p < .0083). To address this, we have opted to apply the False Discovery Rate (FDR) correction using the Benjamini-Hochberg (BH) method. This approach strikes a balance between controlling for false positives across multiple comparisons and avoiding overly conservative outcomes.
Please refer to the revised results section for details on the p-value adjustments. We believe this modification improves the statistical rigor of our analysis while maintaining clarity and validity.
- where did you get the effect size from, line 163? This is a rather big effect, please justify it. Also; is iths ifr a 2x2 design? Or a 2x3x3 design? Furthermore, you run multiple analysis, and it should be based on the most complex one, here the 2x2x3x3
Response: Thank you for your insightful comments. We appreciate your concerns regarding the effect size.The selection of f=0.25 was based on previous literature (Huang et al., 2019; Pauligk et al., 2019; Sun et al., 2020).We acknowledge your observation about the effect size being potentially large. Based on the effect size reference values (Cohen, 1988), we have chosen f=0.15 as our effect size estimate.
In line with your suggestion, we have conducted our statistical analysis using a 2x2x3x3 design. The two teacher types serve as the between-subjects variable, while the two action types/text prompts, along with the three brain hemispheres and three brain regions, are treated as within-subjects variables. The calculations indicate that a total of at least 26 participants are needed, with an average of 13 participants per group (with f=0.15,α=0.05,and power=0.8).
We hope this revised approach addresses your concerns. Thank you again for your valuable feedback.
Please see the method section, specifically lines 168-175 highlighted in blue.
“The study employed a 2 (expertise level: expert vs. novice) × 2 (text prompt: how vs. why) × 2 (action type: normative vs. non-normative) mixed experimental design for the behavioral data, with expertise level as a between-subjects variable and text prompt and action type as within-subjects variables. For the EEG data analysis, we used a 2 (expertise level) × 2 (text prompt) × 3 (brain region) × 3 (hemisphere) design, reflecting the full complexity of the experiment.
Initially, 46 teachers were recruited, but 8 participants were excluded due to large artifacts in their EEG data, resulting in a final sample of 38 participants. Power calculations using G*Power software, with a more conservative effect size of f = 0.15, indicated a minimum required sample size of 26 participants (13 per group) to achieve adequate power (α = 0.05, power = 0.80). Our final sample of 38 participants provides sufficient power for both behavioral and EEG analyses.”.
Reference:
Huang, L., Yang, X., Huang, Z., Wang, Y. Brain spatio-temporal dynamics of understanding kind versus hostile intentions based on dyadic body movements. Acta Psychologica Sinica. 2019, 51(5), 557–570. https://doi.org/10.3724/SP.J.1041.2019.00557
Pauligk, S., Kotz, S.A. & Kanske, P. Differential Impact of Emotion on Semantic Processing of Abstract and Concrete Words: ERP and fMRI Evidence. Sci Rep. 2019, 9, 14439. https://doi.org/10.1038/s41598-019-50755-3
Sun, B., Xiao, W., Feng, X., Shao, Y., Zhang, W., & Li, W. Behavioral and brain synchronization differences between expert and novice teachers when collaborating with students. Brain and Cognition. 2020, 139, 105513. https://doi.org/10.1016/j.bandc.2019.105513
- analysis: were trials with error excluded from ERP / EEG analysis?
Response: In our analysis, we excluded trials with errors from the ERP/EEG data to maintain the quality and accuracy of the results. This exclusion allows for a more precise examination of the cognitive processes involved in the tasks, free from the influence of incorrect responses. Additionally, we have added the following statement in the manuscript: “For EEG data, to avoid false positive results, we analyzed the EEG data in two parts, and we excluded trials with errors from the EEG data to maintain the quality and accuracy of the results.”. Please see lines 297-298 highlighted in blue.
- discussion: larger N250 in novice teachers may indicate that they are less familiar with the behaviour, hence larger prediction error or arousal of seeing that, that fits with the predictive processing account of the brain
Response: Thank you for your valuable feedback regarding the discussion of the N250 findings. In response to your comment, we have expanded the discussion to include the interpretation that the larger N250 in novice teachers may indeed indicate their lesser familiarity with the normative behaviors being assessed. This observation aligns with the predictive processing account of the brain, suggesting that the increased N250 reflects greater prediction error or heightened arousal when encountering behaviors that are unfamiliar. Please see the discussion section, specifically lines 519-532 highlighted in blue.
“ERP results indicate that novice teachers exhibited larger N250 amplitudes in response to classroom actions, particularly in the fronto-central regions of the brain. The N250 is linked to the brain's predictive processing system [64], where larger amplitudes are indicative of greater prediction error or heightened arousal when encountering unfamiliar stimuli. Novice teachers, due to their relative inexperience, are more likely to experience discrepancies between expected and actual classroom actions, resulting in the larger N250 amplitudes. This supports the notion that novice teachers may face greater cognitive challenges in interpreting students’ behaviors, as they must allocate more cognitive resources toward making sense of classroom interactions. In contrast, normative actions—those that align with teachers’ expectations—elicited more negative N250 amplitudes, indicating that familiar behaviors are processed with less cognitive effort. This mirrors findings from studies on face recognition [64-66], where familiar faces elicit larger N250 responses due to their ability to attract and hold attention more effectively.”.
Reference:
Hutchinson, J. B., & Barrett, L. F. (2019). The Power of Predictions: An Emerging Paradigm for Psychological Research. Current Directions in Psychological Science, 28(3), 280-291. https://doi.org/10.1177/0963721419831992
- the discussion is devoid of the how/why difference?
Response: Thank you for your insightful feedback regarding the discussion section. We appreciate your observation that the how/why difference was not adequately addressed. In response, we have expanded the discussion to clarify the distinctions between the "how" and "why" prompts.
We have highlighted how the "how" prompt typically requires a more straightforward recognition of actions, leading to quicker response times and higher accuracy in judgments. Conversely, the "why" prompt necessitates a deeper cognitive processing level, as it involves understanding the underlying intentions behind actions. This distinction is essential in interpreting the EEG results, particularly the observed differences in P300 amplitudes between the two conditions. Please see the discussion section, specifically lines 554-569 highlighted in blue.
“The behavioral results indicate that teachers' response times to "how" prompts are significantly faster than those for "why" prompts, with higher accuracy in their judgments regarding students' action intentions at the "how" level. This finding suggests that lower-level prompts are easier to process. In contrast, the EEG results reveal that the P300 amplitude induced by "why" prompts is significantly greater than that induced by "how" prompts. Research indicated that understanding the intentions behind "why" questions primarily activates brain regions associated with theory of mind, such as the ventromedial prefrontal cortex [68]. This processing is influenced by cognitive load, whereas understanding "how" predominantly engages areas related to the mirror neuron system, such as the inferior parietal lobule, which is less affected by cognitive demands. In this experiment, understanding "how" requires teachers to recognize students' actions at a basic level, while comprehending "why" students engage in certain behaviors necessitates deeper cognitive processing, involving critical thinking about the underlying motivations for those actions. Consequently, grasping the reasons behind students' behaviors demands additional cognitive effort, thereby consuming more cognitive resources from teachers.”.
Reference:
Spunt, R. P., & Lieberman, M. D. (2013). The busy social brain: Evidence for automaticity and control in the neural systems supporting social cognition and action understanding. Psychological Science, 24(1), 80-86. https://doi.org/10.1177/0956797612450884
- conclusion: "... better understood student's behavioural intentions..." - even if you see a difference in P300 and LPC you cannot claim that. You have not measured it, this is inferred solely from the LPC and P300 but there are alternative explaantions. Maybe yoou can say so if the P300 and LPC is higher in why than how trials of non-normative behaviour, but you will not have the stastical power for a 2x2x2 analysis (expert/novice x how/why x normative/non-normative) and LPC and P300 as outcome.
Response: Thank you for your valuable feedback regarding our conclusion. We have revised the conclusion to clarify that while we observed differences in P300 and LPC amplitudes between expert and novice teachers, we cannot definitively claim that these differences indicate a better understanding of students' behavioral intentions. We acknowledge that these findings are inferred from the ERP data, and alternative explanations may exist.
Specific changes are as follows: “We found that, compared with novices, expert teachers demonstrated different patterns of brain activity, as reflected in the P300 and LPC amplitudes during the late time window. Additionally, novice teachers exhibited a stronger N250 amplitude in the early time window, indicating a potentially heightened response to normative actions. Notably, the N250 amplitude was more pronounced for normative actions, whereas the P300 and LPC amplitudes were significantly larger for non-normative actions.”. Please see lines 598-604 highlighted in blue.
minor: please report always exact p values, not p<.05 but p = .246, also the significant p-value should be written out (unless smaller than .001)
Response: Thank you for your feedback regarding the reporting of p-values. In our revised manuscript, we have made the necessary changes to report all p-values in full, rather than using less specific notations such as "p < .05.". For instance, we have specified p = .022, and we have ensured that all significant p-values are clearly reported unless they are smaller than .001.
line 178 what / where is "normal university"?
Response: Thank you for pointing out the term "normal university" in line 178. We understand that this terminology may not be familiar to all readers. In the context of our study, "normal university" refers to institutions that specifically focus on training teachers and providing education-related programs, often referred to as "teacher training colleges" or "normal schools" in some regions.
please expand the legend for figure 2. normative /non-normative action is a photo? how/why is separate trials or always shown both? (from later mentioned text it seems it is separate trials)
Response: Thank you for your insightful feedback regarding the legend for Figure 2. We appreciate your suggestion to clarify the details related to normative and non-normative actions, as well as the presentation of the how/why conditions. Please see specifically lines 263-269 highlighted in blue.
Revised Legend for Figure 2: “Figure 2. Experimental procedure used in the study. Each trial consisted of a specific sequence: a fixation point was presented for 500 ms, followed by either a "how" or "why" text prompt, which was displayed for 1500 ms in a random order across trials. Subsequently, photographs of students engaged in normative and non-normative actions were presented for 2000 ms. After viewing the photographs, participants made a key press judgment regarding the actions and then rated the comprehensibility of both the text prompt and the student photographs.”
We hope this expansion provides clearer context for Figure 2 and enhances the reader's understanding of the experimental design.
line 277: tabel should be table
Response: Thank you for pointing out the typo in line 277. We have corrected "tabel" to "table" in the manuscript to ensure clarity and accuracy. We appreciate your careful review. The revised sentence is as follows: “The independent samples t-test results showed significant differences between the expert and novice teachers in the three dimensions of the IRI-C scale (Table 1)”
line 291 space issue .001(Table 2) should be .001 (Table 2)
Response: Thank you for bringing the spacing issue in line 291 to our attention. We have corrected it to ensure the formatting now reads ".001 (Table 2)" instead of ".001(Table 2)," enhancing clarity. The revised sentence is as follows:“The interaction between expertise level and test time was not significant, F(1, 36) = 0.019, p = 0.890, ηp² = 0.001 (Table 2).”
line 442/443 / 455: 2x simple, delete one instance
Response: Thank you for your careful review and for pointing out the redundancy in the text. We have deleted the repeated instance of "2x simple" in lines 442, 443, and 455 to enhance clarity and conciseness. The revised text now reads: “Simple effects analyses showed that…”. We appreciate your attention to detail, which has contributed to improving the quality of our manuscript.
figure 4 and 5 are not shwon properly, the legend is out of range
Response: Thank you for your observation regarding figures 4 and 5. We have revised the figures to ensure that they are displayed properly and that the legends are within the appropriate range. This adjustment should enhance the clarity and readability of the figures.
line 473: "... larger degree of late activation" should that be "larger degree at late activation"?
Response: Thank you for pointing out the phrasing issue. We have revised the sentence for clarity. It now reads, " indicating that the teacher understands the student's intention to a larger degree at late activation. " to accurately convey the intended meaning.
line 475 space issue the270
Response: Thank you for bringing the spacing issue to our attention. We have corrected the formatting to ensure proper spacing around "the 270" and have conducted a thorough review of the manuscript to ensure that there are no additional spacing issues throughout the text.
there are general space issues for F() i.e. there should be no space between F and bracket
Response: Thank you for highlighting the spacing issues related to the F-statistic notation. We have thoroughly reviewed the manuscript and corrected all instances to ensure there is no space between "F" and the opening bracket in all relevant sections. Additionally, we have marked all modified instances of "F" in blue for your convenience. We appreciate your attention to detail, which has helped us improve the clarity and presentation of our manuscript.
line 484ff: "... show that ... novel teachers there were no ..." grammatically wrong. replace "there were no" with "had no"
Response: Thank you for your observation regarding the grammatical error. We have revised the text as follows: "behavioral results show that expert teachers and novice teachers had no significant differences in reaction time and accuracy rate". This change improves the grammatical accuracy of the sentence.
line 516: trails should be trials
overall an interesting read, thanks.
Response: Thank you for pointing out the typo. We have corrected "trails" to "trials" in line 516 to ensure accuracy. “subjects need to superimpose multiple sets of blocks and multiple trials in a short period of time.”.
Comments on the Quality of English Language
some grammatically odd construct
e.g. line 433ff "... teacher's expertise levels have differences in understanding students' classroom action intention,..."
could be "yielded differences" but since there is no group differences at the behavioural level, the entire construct doesn't work
there are similar instances throught the manuscript
another example on line 349ff: ...teachers understand that the N250 amplitude ..." I don't think teachers understand the N250 amplitude, please rewrite
Response: Thank you for your constructive feedback regarding the quality of the English language in our manuscript. We sincerely apologize for the grammatical errors you identified and appreciate your guidance in improving our writing.
In response to your comments, we have revised the phrase in line 433 to "yielded differences" to better reflect the intended meaning. Additionally, we have thoroughly reviewed the manuscript to correct similar instances of awkward phrasing and ensure clarity throughout. Please see 445-446 line, “Teacher’ expertise levels yielded differences in understanding students’ classroom action intention, the N250 amplitude of novice teachers was more negative than that of expert teachers”.
Regarding your second comment about the N250 amplitude, we have taken into account both your feedback and that of Reviewer 2. As a result, we have rewritten the relevant section in the results to clarify that the interpretation of the N250 amplitude is based on existing research rather than implying that teachers understand this neural measure directly.
Please refer to the revised results section for these updates, specifically lines 361-375 highlighted in blue.
We appreciate your insights and believe these changes enhance the clarity and quality of our manuscript. Thank you once again for your valuable suggestions. We look forward to any further comments you may have.

Reviewer 2 Report
Comments and Suggestions for Authors
Expert and Novice Teachers’ Cognitive Neural Differences in Understanding Students’ Classroom Action Intentions
In the INTRODUCTION, the authors need to add a more specific definition of expertise. I know that there is not a unique definition of this construct, but the authors need to define it. Moreover, The authors introduced the expertise in the chess. However, from a conceptual point of view, chess expertise is different from specific training in education sciences ( or to become a teacher). This needs to be highlighted in the text.
Moreover, the embodied cognition and the link with expertise are not completely clear. The authors need to clarify better the role of embodied cognition in the “expertise” from a general point of view, then highlight the peculiarity in the teaching expertise.
Similarly, the authors explained embodied cognition using fMRI studies. However, this is an EEG study, and my advice is to prioritize the EEG results instead of those obtained from fMRI.
The hypotheses are clear.
Methods: The procedure and the task are well-described, as well as the participants. However, it is not clear what are the principal characteristics that distinguish EXPERTS from NOVICES. The criteria seem to be used by authorities to assess the outcome and the results obtained by each teacher, as well as in other countries. However, more info is needed. Please, add the criteria used by authorities to assess the expertise of the teachers.
This statement is not clear: “ According to previous action intention understanding research [28, 34, 48, 53], an 255 observation and analysis of the total average graph, nine electrode sites were selected for 256 analysis: F3/Fz/F4, C3/Cz/C4, P3/Pz/P4.” Please clarify or rewrite it. Moreover, it is not clear if the RM ANOVA is a repeated measure or a mixed-design ANOVA. Please clarify.
Similarly, “and Bonferroni corrections were used for multiple comparisons”, please indicate the post-hoc comparison used (ANOVA).
Results: EEG (ERPs) results are detailed described and I recommend summarizing them. The authors used expressions such as “four-factor analysis” This is not clear, please clarify. I suppose that it is referred to the ANOVA.
DISCUSSION: The frontal lobe is important not only for teachers, in terms of the theory of mind and executive functions. These sentences need to be revised. As a general recommendation, I suggest to remove the sub-section and try to integrate the discussion of the results without fragmenting it.
Author Response
In the INTRODUCTION, the authors need to add a more specific definition of expertise. I know that there is not a unique definition of this construct, but the authors need to define it. Moreover, The authors introduced the expertise in the chess. However, from a conceptual point of view, chess expertise is different from specific training in education sciences (or to become a teacher). This needs to be highlighted in the text.
Response: Thank you for your insightful feedback regarding the definition of expertise in the introduction. We agree that a clearer definition is necessary, given the varied interpretations of this construct across domains. In response, we have provided a more specific definition of teaching expertise, which encompasses not only subject matter knowledge but also the ability to manage classroom dynamics, recognize student behavior patterns, and make real-time decisions to enhance learning. This definition has been added to the introduction for better clarity.
Furthermore, we have revised the text to explicitly highlight the differences between chess expertise and teaching expertise. While expertise across fields shares common cognitive components—such as pattern recognition and decision-making—teaching expertise involves the additional complexity of managing live, interactive classroom environments. Please see the introduction section, specifically lines 42-61 highlighted in blue.
“Expertise, broadly defined as the proficiency or skill developed through extensive experience or training in a particular domain, varies significantly across fields [11, 12]. In education, teaching expertise goes beyond mere subject knowledge; it encompasses the ability to anticipate, understand, and respond to the dynamic and often unpredictable classroom environment. This includes recognizing patterns in student behaviors and making timely, effective decisions to facilitate learning [13]. Compared to novices, experts demonstrate superior cognitive processing when handling complex information within their area of expertise. They interpret and respond to classroom situations with greater speed and accuracy, relying on their advanced cognitive frameworks and experience. For instance, research in other domains, such as chess, has shown that professional players employ more sophisticated visual search strategies and exhibit enhanced pattern recognition abilities [11-12]. Advanced chess players also show heightened alpha EEG (electroencephalogram) activity during complex tasks, compared to novices [14]. Similarly, in tennis, experts display faster and stronger activation of ERP (event-related potential) components during motion perception tasks [15]. However, the cognitive demands of teaching involve additional layers of complexity, including real-time decision-making, managing live interactions, and responding to the continuous flow of classroom stimuli.
In the field of teacher education, there are obvious differences between experts and novice teachers in their cognitive processing of classroom teaching situation information [1, 16-19].”.
New Reference:
- Pelgrim, E., Hissink, E., Bus, L., van der Schaaf, M., Nieuwenhuis, L., van Tartwijk, J., & Kuijer-Siebelink, W. Professionals' adaptive expertise and adaptive performance in educational and workplace settings: An overview of reviews. Advances in Health Sciences Education: Theory and Practice. 2022, 27(5), 1245–1263. https://doi.org/10.1007/s10459-022-10190-y
- van Tartwijk, J., van Dijk, E. E., Geertsema, J., Kluijtmans, M., & van der Schaaf, M. Teacher expertise and how it develops during teachers’ professional lives. International Encyclopedia of Education (Fourth Edition), 2023, 170–179. Elsevier.
- Wieser, C. Teaching expertise in higher education: Constructive alignment and how an experienced university teacher maintained constructive alignment in practice. In C. Wieser & A. Pilch Ortega (Eds.), Ethnography in higher education: Doing higher education (pp. [insert page range]). 2020, Springer VS. https://doi.org/10.1007/978-3-658-30381-5_3
Moreover, the embodied cognition and the link with expertise are not completely clear. The authors need to clarify better the role of embodied cognition in the “expertise” from a general point of view, then highlight the peculiarity in the teaching expertise.
Similarly, the authors explained embodied cognition using fMRI studies. However, this is an EEG study, and my advice is to prioritize the EEG results instead of those obtained from fMRI.
The hypotheses are clear.
Response: Thank you for your feedback on the clarity of embodied cognition and its link to expertise. We have enhanced our discussion by providing a clearer definition of embodied cognition, emphasizing its role in expertise as both mental knowledge and physical engagement. We highlighted how teachers utilize embodied cognition to interpret students' non-verbal cues and manage classroom dynamics effectively. Additionally, we prioritized EEG results in our manuscript, focusing on findings from EEG studies related to event-related potential (ERP) components. Please see the revised introduction section, specifically lines 103-117 highlighted in blue.
“This process is particularly relevant in teaching expertise, where teachers utilize their own pedagogical experiences to interpret students’ behaviors, allowing for swift, accurate decision-making in real-time classroom environments [32]. Embodied cognition plays a critical role in expertise development, as experts—unlike novices—can draw upon both sensory input and their extensive experience to create richer cognitive representations of observed actions. In teaching, this enables expert teachers to not only interpret students’ behaviors but also to predict potential outcomes and guide their instructional responses accordingly [33]. This highlights the unique demands of teaching expertise, which involves managing live, dynamic interactions and interpreting a continuous flow of stimuli in real-time.
From a neurological perspective, the inference of others' intentions within the framework of embodied cognition can be elucidated through millisecond-level event-related potential (ERP) techniques, which reveal the temporal dynamics of intention understanding. Research shows that the main ERP components related to intention understanding include the N250, P300, and late positive component (LPC) [34].”.
New Reference
- Kosmas, P., & Zaphiris, P. Embodied cognition and its implications in education: An overview of recent literature. Interna-tional Journal of Educational and Pedagogical Sciences. 2018, 12(7), 946-950. https://doi.org/10.1999/1307-6892/10009334
- Kiefer, M., & Trumpp, N. M. Embodiment theory and education: The foundations of cognition in perception and action. Trends in Neuroscience and Education. 2012, 1(1), 15–20. https://doi.org/10.1016/j.tine.2012.07.002
Methods: The procedure and the task are well-described, as well as the participants. However, it is not clear what are the principal characteristics that distinguish EXPERTS from NOVICES. The criteria seem to be used by authorities to assess the outcome and the results obtained by each teacher, as well as in other countries. However, more info is needed. Please, add the criteria used by authorities to assess the expertise of the teachers.
Response: Thank you for your valuable feedback regarding the criteria distinguishing expert and novice teachers in our study. In response, we have clarified the categorization of teachers based on established criteria utilized by educational authorities. Please see the revised method section, specifically lines 187-204 highlighted in blue.
“In this study, teachers were categorized as either experts or novices based on several established criteria utilized by educational authorities. Expert teachers typically exhibit the following characteristics: Teaching Experience: A minimum of ten years of teaching experience in their subject area. Degree Requirements: Advanced degrees (Master's or higher) in education or related fields. Professional Development: Active participation in ongoing professional development programs. Effectiveness in Teaching: Demonstrated effectiveness in enhancing student learning outcomes, often evaluated through standardized test scores and qualitative assessments from peers and supervisors.
Novice teachers were selected according to the screening criteria established by existing national and international studies [45-46]. This group comprised senior college students and newly qualified teachers. Senior college students were enrolled in education-focused universities, had undergone teacher training, and completed over three months of practical experience in schools. Newly qualified teachers were classified as novices based on established criteria set by educational authorities, indicating they had been in the profession for less than five years. The senior students primarily attended normal universities, while the novice teachers were employed in various schools across different cities in Zhejiang Province.”.
This statement is not clear: “ According to previous action intention understanding research [28, 34, 48, 53], an 255 observation and analysis of the total average graph, nine electrode sites were selected for 256 analysis: F3/Fz/F4, C3/Cz/C4, P3/Pz/P4.” Please clarify or rewrite it. Moreover, it is not clear if the RM ANOVA is a repeated measure or a mixed-design ANOVA. Please clarify.
Response: Thank you for highlighting these points of confusion. We appreciate your attention to clarity in our manuscript. To clarify the statement regarding electrode sites, we have rewritten it as follows, lines 285-288: "Based on previous research on action intention understanding [28, 34, 48, 53], we conducted an observation and analysis of the average EEG data across all trials. From this analysis, we selected nine electrode sites for further examination: F3, Fz, F4, C3, Cz, C4, P3, Pz, and P4.".
Regarding the analysis method, we would like to clarify that the RM ANOVA employed in our study is a repeated measures ANOVA. This approach was used to account for the within-subjects nature of our experimental design, where the same participants were measured across different conditions. Please see revised method part, lines 293-296: “We utilized SPSS 22.0 to perform repeated measures ANOVAs with a 2 (expertise level: expert vs. novice) × 2 (text prompt: how vs. why) × 2 (action type: normative vs. non-normative) design. The dependent variables included response time, accuracy, and comprehensibility for the behavioral data”.
We hope these revisions enhance the clarity of our methods and analysis.
Similarly, “and Bonferroni corrections were used for multiple comparisons”, please indicate the post-hoc comparison used (ANOVA).
Response: Thank you for your valuable feedback regarding the clarity of our post-hoc comparisons. We apologize for any confusion in our previous description. In response to your and Reviewer 1's comments, we have corrected the method for post-hoc comparisons to utilize the False Discovery Rate (FDR) adjustment. Please see revised method part, lines 304-307: “The Greenhouse-Geisser correction was applied to adjust the degrees of freedom when the assumptions of sphericity were violated. Post-hoc comparisons were conducted using ANOVA with a False Discovery Rate (FDR) adjustment based on the adjusted p-values.”. Thank you for your guidance on this matter.
Results: EEG (ERPs) results are detailed described and I recommend summarizing them. The authors used expressions such as “four-factor analysis” This is not clear, please clarify. I suppose that it is referred to the ANOVA.
Response: Thank you for your valuable feedback regarding the EEG (ERPs) results and the terminology used in our manuscript. In response to your suggestion, we have summarized the EEG results to enhance clarity and conciseness. The revised summary captures the key findings while providing essential context. Please refer to the revised results section, lines 359-472.
We also apologize for any confusion caused by the term "four-factor analysis." To avoid ambiguity, we have revised the text to explicitly state "four-factor ANOVA." The relevant sections have been updated to reflect this change: 3.3.1 The First Four-Factor ANOVA Analysis and 3.3.2 The Second Four-Factor ANOVA Analysis.
We appreciate your suggestions and believe these changes will improve the clarity and effectiveness of our results section.
DISCUSSION: The frontal lobe is important not only for teachers, in terms of the theory of mind and executive functions. These sentences need to be revised. As a general recommendation, I suggest to remove the sub-section and try to integrate the discussion of the results without fragmenting it.
Response: Thank you for your valuable feedback on our manuscript. We appreciate your suggestion to revise the statements regarding the frontal lobe's significance for teachers, particularly in relation to the theory of mind and executive functions. In response, we have revised the relevant sentences to clarify that the frontal lobe plays a crucial role in cognitive processes not only for teachers but also in broader educational contexts. This revision emphasizes the importance of the frontal lobe in understanding and managing social interactions and cognitive tasks across various educational roles. Please refer to the revised discussion section for the updated content, specifically lines 488-569 highlighted in blue.
We believe these changes enhance the clarity and depth of our findings and better highlight the implications of our research for educational practices. Thank you once again for your thoughtful review. We look forward to your further comments.

Round 2
Reviewer 2 Report
Comments and Suggestions for Authors
The authors have addressed my concerns. However, in the results, I suggest removing "3.3.1 The first four-factor ANOVA analysis " and specifying the paragraph on the basis of the principal objective (i.e. highlighting "action" or "text"). According to me, this can facilitate the reading of the results.
Author Response
Comments 1: The authors have addressed my concerns. However, in the results, I suggest
removing "3.3.1 The first four-factor ANOVA analysis " and specifying the paragraph on the
basis of the principal objective (i.e. highlighting "action" or "text"). According to me, this can
facilitate the reading of the results.
Response 1: Thank you very much for your feedback and thoughtful suggestions. We have
carefully considered your recommendation to improve the readability of the results section.
In response, we have removed the section titled "3.3.1 The first four-factor ANOVA analysis"
and reorganized the results according to the principal objective, specifically highlighting
"action" and "text" where relevant. We believe this adjustment will enhance the clarity and
flow of the results, making it easier for readers to follow the main findings. Please see the
revised results section with blue highlight, where these changes have been implemented.
“To examine the effects of expertise level and action type on EEG activity, we conducted a 2×2×3×3
repeated measures analysis of variance (ANOVA) on the N250, P300, and LPC components. Expertise
level was treated as the between-subjects factor, while action type, brain region, and hemisphere were
included as within-subject factors. The specific results are as follows.
3.3.1. N250 component 3.3.2. P300 component 3.3.3. Late Positive Component
To examine the effects of expertise level and text prompts on EEG activity, we conducted a second
2×2×3×3 repeated measures analysis of variance (ANOVA) on the N250, P300, and LPC components.
Expertise level was treated as the between-subjects factor, while text prompts, brain region, and
hemisphere were included as within-subject factors. The specific results are as follows.
3.3.4. N250 (A Second Repeated Measures ANOVA) 3.3.5. P300 (A Second Repeated Measures ANOVA)
3.3.6. LPC (A Second Repeated Measures ANOVA)”
